# Chemical Hypoxia Induces Pyroptosis in Neuronal Cells by Caspase-Dependent Gasdermin Activation

**DOI:** 10.3390/ijms25042185

**Published:** 2024-02-11

**Authors:** Chan Ho Park, Jun Young Park, Won Gil Cho

**Affiliations:** 1Department of Anatomy, Yonsei University Wonju College of Medicine, 20 Ilsan-ro, Wonju 26426, Republic of Korea; wktlr8@naver.com; 2Department of Nuclear Medicine, Severance Hospital, Yonsei University College of Medicine, 50-1 Yonsei-ro, Seodaemun-gu, Seoul 03722, Republic of Korea; abies60@naver.com

**Keywords:** hypoxia, CoCl_2_, reactive oxygen species, gasdermin, pyroptosis

## Abstract

Hypoxia-induced neuronal death is a major cause of neurodegenerative diseases. Pyroptosis is a type of inflammatory programmed cell death mediated by elevated intracellular levels of reactive oxygen species (ROS). Therefore, we hypothesized that hypoxia-induced ROS may trigger pyroptosis via caspase-dependent gasdermin (GSDM) activation in neuronal cells. To test this, we exposed SH-SY5Y neuronal cells to cobalt chloride (CoCl_2_) to trigger hypoxia and then evaluated the cellular and molecular responses to hypoxic conditions. Our data revealed that CoCl_2_ induced cell growth inhibition and the expression of hypoxia-inducible factor-1α in SH-SY5Y cells. Exposure to CoCl_2_ elicits excessive accumulation of cytosolic and mitochondrial ROS in SH-SY5Y cells. CoCl_2_-induced hypoxia not only activated the intrinsic (caspases-3, -7, and -9) apoptotic pathway but also induced caspase-3/GSDME-dependent and NLRP3/caspase-1/GSDMD-mediated pyroptosis in SH-SY5Y cells. Importantly, inhibition of caspase-3 and -1 using selective inhibitors ameliorated pyroptotic cell death and downregulated GSDM protein expression. Additionally, treatment with a ROS scavenger significantly suppressed caspase- and pyroptosis-related proteins in CoCl_2_-treated SH-SY5Y cells. Our findings indicate that hypoxia-mediated ROS production plays an important role in the activation of both apoptosis and pyroptosis in SH-SY5Y neuronal cells, thus providing a potential therapeutic strategy for hypoxia-related neurological diseases.

## 1. Introduction

Hypoxia is a condition in which the supply of oxygen is insufficient to maintain normal cellular function in cells, tissues, and organs [1]. The brain consumes approximately 20 percent of the total oxygen supply to generate adenosine triphosphate (ATP) molecules that are essential for cell integrity, brain tissue viability, and electrophysiological activity [2]. ATP is synthesized via the oxidative phosphorylation pathway in mitochondria under aerobic conditions [3]. However, hypoxic conditions diminish mitochondrial ATP production, thus resulting in irreversible cellular dysfunction, including disturbances in ion channel homeostasis, membrane damage, impaired mitochondrial metabolism, and reactive oxygen species (ROS) production [4,5]. Studies have demonstrated that hypoxia is pivotal in triggering neuronal cell death that can lead to neurodegenerative disorders, including Alzheimer’s disease [6], Parkinson’s disease [7], multiple sclerosis [8], Huntington’s disease [9], and amyotrophic lateral sclerosis [10].

Cellular ROS are primarily generated by mitochondria during cellular respiration in the electron transport chain (ETC) where electrons leak and directly interact with oxygen to form superoxide anions, hydroxyl ions, and hydrogen peroxide in mitochondrial complexes I and III [11]. However, the net production of ROS from mitochondria can be markedly elevated under various pathological conditions, including hypoxia, ischemia, reperfusion, inflammation, and genetic defects in the ETC [12,13]. Hypoxia induces excessive ROS production through inefficient electron transport to the ETC in the mitochondria due to the lack of electron acceptor oxygen [14]. In Alzheimer’s disease, the decrease in cerebral blood flow causes cerebral hypoxia that enhances neuroinflammation, amyloid-β (Aβ) production, tau hyperphosphorylation, and mitochondrial dysfunction [15]. Moreover, the Aβ aggregates activate the NADPH oxidase 2 that is highly expressed in microglial cells, thus resulting in an increase in ROS release [16]. In Parkinson’s disease, mitochondrial complex I deficiency is abundantly observed in dopaminergic neurons of the substantia nigra [17].

Hypoxia-inducible factor-1 (HIF-1), a heterodimeric transcription factor, plays a central role in the cellular response to low oxygen concentrations [18]. HIF-1 consists of the oxygen-sensitive HIF-1α and the constitutively stable HIF-1β, both of which are basic helix-loop-helix proteins containing a PAS domain that is responsible for dimerization and DNA binding [19]. Studies have demonstrated that hypoxia enhances the levels of HIF-1α, ultimately leading to translocation to the nucleus, dimerization with HIF-1β, and binding to hypoxia-response elements for the transcription of target genes to adapt to a low oxygen environment [20]. In hypoxic conditions, overexpressed HIF-1α triggers the transcription of the β-secretase 1 gene and directly interacts with the γ-secretase complex. A significant increase in β- and γ-secretase activities promotes the cleavage of Aβ precursor protein and the deposition of extracellular Aβ plaques in Alzheimer’s disease [21].

Cell death is an essential process for organism development, maintenance of tissue homeostasis, and host defense against pathogens [22]. Cell death can typically be classified as regulated or non-regulated based on signal dependency. Unregulated cell death is caused by an unexpected cell injury. In contrast, regulated cell death, also known as programmed cell death, is regulated by intracellular signaling pathways [23]. Pyroptosis is a lytic form of programmed cell death characterized by the formation of pores in the plasma membrane [24]. Gasdermin (GSDM) consists of functional N-terminal and inhibitory C-terminal domains connected by a flexible interdomain linker. In response to diverse damaging signals, the linker region of gasdermin is cleaved by active caspases or granzymes, and the N-terminal fragments are translocated to the plasma membrane to form pores [25]. Recent studies have reported that pyroptosis can be induced by hypoxia [26,27]. Under hypoxic conditions, PD-L1 physically interacts with p-Stat3 and translocates to the nucleus, ultimately enhancing gasdermin C (GSDMC) gene and protein expression. Caspase-8 activated by macrophage-derived TNFα cleaves GSDMC, thus resulting in a switch from apoptosis to pyroptosis in cancer cells [28]. Hypoxia induces ROS overproduction and promotes the NF-κB/HIF-1α signaling pathways, and the enhanced expression of cleaved caspase-1 and gasdermin D (GSDMD) further promotes pyroptosis in CoCl_2_ myoblasts [29]. Moreover, chronic intermittent hypoxia induces GSDMD-mediated pyroptosis via activation of the NLRP3 inflammasome and caspase-1 in renal tubular epithelial cells [30]. However, the molecular mechanisms underlying hypoxia-induced pyroptosis in neurons remain largely unknown. In this study, we investigated the role of hypoxia in neuronal cell pyroptosis and the possible molecular mechanisms underlying pyroptosis under hypoxic conditions.

## 2. Results

### 2.1. Effect of CoCl_2_-Induced Hypoxia on Cell Viability and Morphological Changes

Cobalt chloride (CoCl_2_) is a well-known hypoxia-mimetic chemical compound that induces hypoxia-like conditions in vitro [31]. To investigate whether CoCl_2_ leads to neuronal cytotoxicity, SH-SY5Y neuronal cells were treated with different concentrations of CoCl_2,_ and cell viability was monitored at 24 h. As shown in Figure 1A, CoCl_2_ treatment significantly induced cell death in a dose-dependent manner. In particular, a significant reduction of cell viability was observed after exposure to 100 μM CoCl_2_ compared with that of the control cells. When the concentration of CoCl_2_ reached 200 μM, the cell viability of SH-SY5Y decreased around 50%. Therefore, we selected CoCl_2_ concentrations of 200 μM for subsequent experiments.

The effect of CoCl_2_ in cell proliferation and cell morphology was also observed in a dose-dependent manner using phase-contrast microscopy. As shown in Figure 1B, there was no significant change in SH-SY5Y cell population and cell morphology under 50 µM CoCl_2_ treatment. However, SH-SY5Y cells treated with ≥100 μM CoCl_2_ exhibited a significant decrease in cell numbers and morphological alterations including rounding up, membrane blebbing, cell shrinkage and neurite degeneration. These results indicated that hypoxia may adversely affect neuronal cell viability as well as cell morphology.

### 2.2. Chemical Hypoxia Induces Accumulation of HIF-1α in SH-SY5Y Cells

To examine if CoCl_2_-induced hypoxia affects the protein expression of HIF-1α, we determined the expression levels of HIF-1α protein by Western blot analysis in SH-SY5Y cells treated with different CoCl_2_ concentrations up to 200 μM. Our results demonstrated that HIF-1α expression was absent in the untreated SH-SY5Y cells (Figure 2A). However, CoCl_2_ induced the increase in HIF-1α protein expression in a dose-dependent manner. In particular, the protein expression of HIF-1α was significantly increased at the 200 µM concentration of CoCl_2_ compared to the levels in response to the other CoCl_2_ doses (*p* < 0.001) (Figure 2B). These results suggested that CoCl_2_ mimicked hypoxia in SH-SY5Y cells.

### 2.3. Chemical Hypoxia Enhances Cytosolic and Mitochondrial ROS

Hypoxic conditions promote increased intracellular oxidative stress, particularly in the form of ROS [32]. To explore if CoCl_2_ treatment induces oxidative stress, we assessed the cytosolic and mitochondrial ROS generation in CoCl_2_-exposed SH-SY5Y cells (Figure 2C). Cytosolic ROS production induced by CoCl_2_ was determined by 2′-7′dichlorofluorescin diacetate (DCFH-DA) assay. Results indicate that the cytosolic ROS level was significantly increased in 100 or 200 μM CoCl_2_-treated SH-SY5Y cells. As ROS generated by mitochondrial Complex III under hypoxia stabilizes the HIF-1α protein [33], mitochondrial superoxide anion (O_2_^−^) levels were determined using MitoSOX. In agreement with changes in intracellular ROS levels, treatment with 200 µM CoCl_2_ markedly increased mitochondrial ROS levels compared to normoxic controls (Figure 2D). These results indicate that CoCl_2_-induced hypoxia increased cytosolic and mitochondrial superoxide generation in SH-SY5Y cells.

### 2.4. Chemical Hypoxia Triggers Intrinsic Apoptotic Pathways in SH-SY5Y Cells

Oxidative stress plays a crucial role in initiating apoptosis [34]. To evaluate the effect of CoCl_2_ on the B cell lymphoma-2 (Bcl-2) family in SH-SY5Y cells, Western blotting was performed to detect the expression of Bax and Bcl-2 proteins and the Bax/Bcl-2 ratio. The results revealed that treatment with CoCl_2_ significantly increased the pro-apoptotic Bax protein level in 100 or 200 μM CoCl_2_-treated SH-SY5Y cells, while levels of antiapoptotic Bcl-2 protein were markedly decreased at 200 μM CoCl_2_ compared to levels in normoxic cells (Figure 3A). The Bax/Bcl-2 ratio was significantly increased following treatment with 100 or 200 µM CoCl_2_ (*p* < 0.05), and this correlated with intracellular ROS generation (Figure 3B).

Hypoxia-induced ROS can activate caspase-9, thus leading to increased apoptosis [35]. Western blotting results demonstrated that the apoptotic initiator caspase-9 was significantly activated following treatment with 100 or 200 μM CoCl_2_ (Figure 3C). Activated caspase-9 can directly cleave the effectors caspase-3 and caspase-7 [36]. Consistent with the activation of caspase-9, the protein levels of cleaved caspase-3 and cleaved caspase-7 were elevated in a dose-dependent manner (Figure 3D,E). Additionally, cleavage of poly-(ADP-ribose)polymerase (PARP), a major substrate of caspase-3 and caspase-7, also increased significantly at 200 μM CoCl_2_ compared to that in the control cells (*p* < 0.001) (Figure 3F). These results suggest that CoCl_2_-induced hypoxic conditions can induce classical caspase-dependent apoptosis in SH-SY5Y cells.

### 2.5. Chemical Hypoxia Activates Caspase-3/GSDME-Mediated Pyroptosis

Recent studies have identified that caspase-3 plays a key executor protein in both apoptotic and pyroptotic pathways [37]. Activated caspase-3 cleaves gasdermin E (GSDME) to generate a N-terminal fragment of GSDME (GSDME-NT) that induces pyroptosis by forming membrane pores [38,39]. To investigate the effect of CoCl_2_-induced hypoxia on the GSDME-mediated pyroptosis in SH-SY5Y cells, we assessed GSDME protein expression by Western blotting (Figure 4A). The results revealed that the level of GSDME-NT was significantly increased when SH-SY5Y cells were treated with 200 μM of CoCl_2_, and this was consistent with the increased levels of cleaved caspase-3 (Figure 4B,C). These results suggested that CoCl_2_ induced GSDME-dependent pyroptosis through caspase-3 activation.

### 2.6. Chemical Hypoxia Causes NLRP3/Caspase-1/GSDMD-Mediated Pyroptosis

The nucleotide-binding oligomerization domain-like receptor family pyrin domain containing 3 (NLRP3) inflammasome, a major mediator of the inflammatory response and GSDMD-mediated pyroptosis, can be activated by diverse stimuli, including hypoxia [40]. Therefore, we explored if CoCl_2_ activated the NLRP3 inflammasome and caused GSDMD cleavage (Figure 5A). Following CoCl_2_ treatment, NLRP3 expression increased in a dose-dependent manner (Figure 5B). The protein expression of cleaved caspase-1 was significantly elevated at 200 μM CoCl_2_ compared to that at other CoCl_2_ doses (*p* < 0.01) (Figure 5C). Furthermore, the expression levels of cleaved N-terminal GSDMD (GSDMD-NT) also exhibited a significant increase in 200 μM CoCl_2_-treated SH-SY5Y cells (*p* < 0.01) (Figure 5D). Consistent with activation of the caspase-1/GSDMD cascade, the expression of interleukin-1β (IL-1β) and IL-18 is also significantly increased at concentrations of 100 or 200 μM CoCl_2_ (*p* < 0.01) (Figure 5E,F). These results suggest that CoCl_2_-induced hypoxia induces NLRP3/caspase-1/GSDMD-mediated pyroptosis and downstream inflammation in SH-SY5Y cells.

### 2.7. Hypoxia-Mediated Pyroptosis Depends on the Caspase Activation

Western blotting demonstrated that CoCl_2_ induced the activation of caspase-1 and -3. As caspase-3 is the key protein involved in GSDME-mediated pyroptosis, SH-SY5Y cells were pre-treated with the caspase-3-specific inhibitor benzyloxycarbonyl-Asp(OMe)-Glu(OMe)-Val-Asp(OMe)-fluoromethylketone (Z-DEVD-FMK) in the presence or absence of CoCl_2_. As presented in Figure 6A, Z-DEVD-FMK significantly restored cell viability in the presence of CoCl_2_ (*p* < 0.001). Western blot analysis demonstrated that Z-DEVD-FMK prevented the activation of caspase-3 and the cleavage of GSDME when compared to CoCl_2_-treated SH-SY5Y cells (Figure 6B). Additionally, the GSDMD-derived caspase-1 inhibitor acetyl-Tyr-Val-Ala-Asp-chloromethylketone (Ac-YVAD-CMK) effectively reduced CoCl_2_-induced cell death and attenuated the expression levels of cleaved caspase-1 and cleaved GSDMD in CoCl_2_-treated SH-SY5Y cells (Figure 6C). These results indicated that both the caspase-3/GSDME and caspase-1/GSDMD pathways contribute to hypoxia-mediated cell death in SH-SY5Y cells.

### 2.8. ROS Activation Contributed to Hypoxia-Mediated Pyroptosis

Recent studies indicated that ROS play important roles in the progression of caspase-mediated pyroptosis [41,42]. The present study demonstrated that CoCl_2_ treatment elevates intracellular ROS levels. To further verify the role of ROS in hypoxia-mediated pyroptosis, we investigated if the ROS scavenger N-acetyl-L-cysteine attenuated the upregulation of pyroptosis-related proteins in SH-SY5Y cells. As presented in Figure 7A, N-acetyl-L-cysteine pretreatment significantly suppressed the expression of cleaved caspase-3 and the cleavage of GSDME in CoCl_2_-treated SH-SY5Y cells. However, there was no significant difference in the expression of cleaved caspase-3 after N-acetyl-L-cysteine treatment in the CoCl_2_-treated cells. Furthermore, N-acetyl-L-cysteine reduced the expression of cleaved caspase-1 protein and the formation of GSDMD-NT compared to levels in CoCl_2_-treated SH-SY5Y cells (Figure 7B). These findings suggest that ROS contributes to the activation of pyroptosis-associated proteins under hypoxic conditions in SH-SY5Y cells.

## 3. Discussion

In the present study, we assessed the effects and mechanisms of hypoxia-mediated pyroptosis in SH-SY5Y cells. We observed that CoCl_2_-induced hypoxia effectively induced ROS accumulation and activation of two cell death pathways (apoptosis and pyroptosis) in SH-SY5Y cells. We also demonstrated that pre-treatment with caspase inhibitors reduced the cleavage of GSDMD and GSDME at the protein level. Additionally, the results from the ROS scavenging assay demonstrated that hypoxia-induced ROS play a key role as executors of pyroptosis.

HIF-1 regulates the gene expression involved in cell survival, angiogenesis, and glucose metabolism in response to cellular oxygen conditions [43]. HIF-1α is unstable and rapidly degraded by the ubiquitin-dependent proteasome system under normoxic conditions [44]. However, under hypoxic conditions HIF-1α is stabilized against degradation after dimerization with HIF-1β [19]. CoCl_2_ artificially induces hypoxia-like conditions by blocking HIF-1α degradation under normoxic conditions [45]. Our results revealed that CoCl_2_ significantly increased cell death as well as the protein expression of HIF-1α in SH-SY5Y neuronal cells. We also confirmed an increase in cytosolic and mitochondrial ROS generation induced by CoCl_2_ in SH-SY5Y cells. In general, ROS are considered major mediators of apoptotic signaling pathways [46]. Apoptosis is executed by caspases, and their activation is mediated by Bcl-2 family proteins, including Bax and Bak [47]. Upon apoptotic stimuli, Bax and Bak are activated and trigger mitochondrial outer membrane permeabilization (MOMP) that leads to the release of proapoptotic proteins, including cytochrome c, from the mitochondrial intermembrane space into the cytoplasm. Cytochrome c interacts with apoptotic protease-activating factor-1 (Apaf-1) to form the Apaf-1 apoptosome that subsequently activates the initiator caspase-9 [48]. Active caspase-9 then cleaves and activates effector caspase-3 and caspase-7. Consistent with these reports, we observed increased levels of Bax protein and cleaved forms of caspase-9, -3, and -7 after stimulation with CoCl_2_ in SH-SY5Y cells.

Pyroptosis is mediated by the gasdermin family members, including GSDMA, GSDMB, GSDMC, GSDMD, GSDME, and GSDMF [25]. Caspase-3 specifically cleaves GSDME to initiate caspase-3-mediated pyroptosis [37]. Recent studies have demonstrated that caspase-3-mediated GSDME pyroptosis actively participates in cancer, acute kidney injury, diabetic nephropathy, and cardiovascular diseases [38,49,50,51]. The present study demonstrates that CoCl_2_ induces an increase in caspase-3 cleavage, thus leading to GSDME activation in SH-SY5Y neuronal cells. However, the caspase-3 inhibitor markedly restored cell viability and inhibited GSDME cleavage in CoCl_2_-treated SH-SY5Y cells.

Hypoxia-induced ROS activate the NLRP3 inflammasome. Previous studies have demonstrated that NLRP3 inflammasome activation contributes to various types of cell death, including apoptosis, necroptosis, ferroptosis, and pyroptosis [52]. In the canonical inflammasome pathway, the NLRP3 inflammasome recruits and activates the caspase-1 that cleaves the cytosolic GSDMD to form a lytic membrane pore and induces secretion of mature IL-1β and IL-18 [29]. In line with these findings, our results demonstrated that CoCl_2_ not only increased the levels of NLRP3, cleaved caspase-1, and cleaved GSDME, but also increased the expression of IL-1β and IL-18 (Figure 5A,B). We also observed that caspase-1 inhibitor pretreatment improved cell proliferation and attenuated the levels of cleaved GSDME protein compared to those in CoCl_2_-treated neuronal cells.

ROS are crucial for tissue homeostasis, cell survival, and cell signaling. However, excessive ROS levels damage cellular components, including membranes, nucleic acids, and organelles [53]. Recent studies have demonstrated that ROS are important mediators of various nervous system diseases. ROS can damage neurons via mitochondrial DNA mutations, MOMP alterations, and homeostasis disruption, ultimately leading to neuronal degeneration and apoptotic cell death [54]. Few studies have associated hypoxia-induced ROS production with pyroptosis in neuronal cells under hypoxic conditions. In the current study, treatment with ROS scavengers decreased the levels of cleaved caspase-3 and GSDME activation and inhibited the expression of cleaved caspase-1 and cleaved GSDMD in CoCl_2_-treated SH-SY5Y neuronal cells. These results demonstrate that hypoxia can induce both apoptosis and pyroptosis or the apoptosis-to-pyroptosis switch through ROS accumulation in SH-SY5Y cells. The limitation of the present study is that the experiments were conducted using only one cell line. Thus, additional comparative studies are needed using normal brain cells, including primary neurons, microglia, and astrocytes. 

In conclusion, we have demonstrated that CoCl_2_-induced hypoxia significantly increased cell death and HIF-1α expression in SH-SY5Y cells, led to the accumulation of cytosolic and mitochondrial ROS, and activated ROS-mediated pyroptosis under hypoxic conditions (Figure 8). These effects were inhibited by caspase-specific inhibitors and attenuated by ROS scavengers in CoCl_2_-treated SH-SY5Y cells. Therefore, we propose that a reduction in ROS levels may be the underlying mechanism of the protective or therapeutic effects of pyroptosis-induced neuronal cell damage during hypoxia.

## 4. Materials and Methods

### 4.1. Materials

Cobalt chloride (Cat# 232696) and N-acetyl-L-cysteine (Cat# A9165) were purchased from Sigma-Aldrich (St. Louis, MO, USA). The caspase inhibitors Ac-YVAD-CMK (Cat# S9727) and Z-DEVD-FMK (Cat# S7312) were purchased from Selleck Chemicals (Houston, TX, USA). The antibodies against HIF-1α (Cat# 36169), Bax (Cat# 2772), B-cell lymphoma-2 (Bcl-2; Cat# 3498), caspase-1 (Cat# 3866), cleaved caspase-1 (Cat# 4199), caspase-3 (Cat# 9662), cleaved caspase-3 (Cat# 9661), cleaved caspase-7 (Cat# 9491), caspase-9 (Cat# 9502), PARP (Cat# 9532), GSDMD (Cat# 97558), IL-1β (Cat# 12242), IL-18 (Cat# 54943), and α-tubulin (Cat# 2125) were purchased from Cell Signaling Technology (Danvers, MA, USA). The antibody against caspase-7 (Cat# sc-28295) was obtained from Santa Cruz Biotechnology (Santa Cruz, CA, USA). The antibodies against NLRP3 (Cat# ab263899) and GSDME (Cat# ab215191) were purchased from Abcam (Cambridge, MA, USA).

### 4.2. Cell Line and Culture

Human neuroblastoma SH-SY5Y cells were purchased from ProCell Life Science and Technology (Wuhan, China). SH-SY5Y cells were grown in Dulbecco’s Modified Eagles Medium (Invitrogen, Carlsbad, CA, USA) supplemented with 10% fetal bovine serum (Gibco, Carlsbad, CA, USA) and 1% penicillin/streptomycin mixture (Sigma-Aldrich) at 37 °C in a humidified incubator containing 5% CO_2_ and 95% air.

### 4.3. Cell Viability Assay

Cell viability was determined using the cell counting kit-8 (CCK-8, DoJinDo Molecular Technology Inc., Kumamoto, Japan) according to the manufacturer’s protocol. Cells were seeded into 96-well plates at 1 × 10^4^ cells per well in 100 µL of standard culture medium and incubated overnight. After the cells were treated with CoCl_2_ at the indicated concentrations, 10 µL of CCK-8 reagent was added to each well and incubated in a tissue culture incubator at 37 °C for an additional hour to enable the reaction of WST-8. The absorbance was measured at 450 nm using a SpectraMax ABS Plus plate reader (Molecular Devices, San Jose, CA, USA). All experiments were independently repeated three times. The viability of CoCl_2_-treated cells is presented as a percentage of that of untreated cells in parallel experiments.

### 4.4. Cell Morphology

SH-SY5Y cells were seeded into 6-well plates at a density of 5 × 10^5^ cells/well in standard culture medium. Cells were treated with CoCl_2_ (0, 50, 100, or 200 µM) for 24 h at 37 °C under a humidified atmosphere of 5% CO_2_ and 95% air. Cell growth, growth conditions, and morphological changes were observed using a phase-contrast microscope (CKX31; Olympus, Tokyo, Japan).

### 4.5. ROS Production Assay

Detection of intracellular and mitochondrial ROS in CoCl_2_-treated SH-SY5Y cells was performed using confocal fluorescence microscopy and flow cytometry. Intracellular ROS was detected using the oxidation-sensitive fluorescent probe DCFH-DA (Cat# ab113851; Abcam). For confocal fluorescence microscopy, SH-SY5Y cells were seeded on glass coverslips and incubated for 24 h and then treated with CoCl_2_ (0, 50, 100, or 200 µM) for 24 h at 37 °C in 5% CO_2_. To determine the amount of intercellular ROS, the culture medium was discarded, and the cells were rinsed with Dulbecco’s phosphate-buffered saline (DPBS) and incubated with 10 µM DCFH-DA in Hank’s balanced salt solution (HBSS) at 37 °C for 30 min in darkness. After washing with HBSS three times, cells were fixed in 4% paraformaldehyde (PFA) for 5 min. Cell nuclei were counterstained with 4′,6-diamidino-2-phenylindole (DAPI) and stained cells were observed under a Zeiss LSM-800 confocal microscope (Zeiss, Oberkochen, Germany). Imaging analysis was performed using ZEN 2010 image software (version 3.3, Zeiss). For flow cytometry analysis, SH-SY5Y cells were seeded in 6-well plates and treated with CoCl_2_ (0, 50, 100, or 200 µM) for 24 h at 37 °C in 5% CO_2_. After washing with HBSS, cells were stained at 37 °C for 20−30 min with 10 µM DCFH-DA. The cells were washed again with DPBS and detached using a 0.25% trypsin/0.53 mM EDTA solution. The cells were then transferred in the FACS tube using growth medium and centrifuged at 200× *g* at 4 °C for 10 min. Samples were resuspended using DPBS and examined by using an FACSAria™ III (Becton Dickinson, Franklin Lakes, NJ, USA), and data were analyzed using FACSDiva (version 6.0, Becton Dickinson).

MitoSOX™ Red superoxide indicator (Cat# M36008; Invitrogen, Waltham, MA, USA) was used to assess mitochondrial ROS levels in CoCl_2_-treated SH-SY5Y cells. The incubation procedure was identical to that used for the DCFH-DA probe. After CoCl_2_ treatment, SH-SY5Y cells were incubated with 5 μM MitoSOX™ Red reagent in HBSS buffer for 10 min at 37 °C in darkness and were then washed twice with PBS and fixed with 4% paraformaldehyde for 15 min. The cell nuclei were labeled with DAPI (1 mg/mL) for 4 min. Fluorescence images were captured using a Zeiss LSM-800 confocal microscope (Zeiss). The fluorescence intensity was measured using a flow cytometer as described above.

### 4.6. Western Blot Analysis

SH-SY5Y cells were harvested and lysed with RIPA lysis buffer (Thermo Fisher Scientific, Waltham, MA, USA) on ice to obtain protein extracts. Lysates were centrifuged at 4 °C for 20 min at 14,000× *g*. The Pierce™ BCA Protein Assay Kit (Cat# 23225, Thermo Fisher Scientific) was used to measure the protein concentration. A total of 20 μg of denatured proteins were separated on 12% sodium dodecyl sulfate polyacrylamide gel electrophoresis and transferred to a polyvinylidene difluoride membrane. After blocking the membrane with 5% skim milk in Tris-buffered saline containing 0.1% Tween-20 (TBST) for 1 h at 37 °C, the membranes were incubated overnight at 4 °C using primary antibodies against HIF-1α (1:1000), Bax (1:1000), B-cell lymphoma-2 (1:1000), caspase-1 (1:1000), cleaved caspase-1 (1:200), caspase-3 (1:1000), cleaved caspase-3 (1:1000), caspase-7 (1:1000), cleaved caspase-7 (1:1000), caspase-9 (1:1000), PARP (1:1000), GSDMD (1:1000), GSDME (Cat# ab215191), IL-1β (1:200), IL-18 (1:200), NLRP3 (1:200), and α-tubulin (1:500). The membranes were washed with TBST and incubated with horseradish peroxidase-conjugated secondary antibodies (1:2500, Cat# SA002-500; GenDEPOT, Barker, TX, USA). The bands were then visualized by the Pierce™ ECL Western Blotting Substrate (Thermo Scientific), and blots were scanned using ChemiDoc XRS+ imaging systems (Bio-Rad Laboratories, Hercules, CA, USA). The intensity of the blot bands was assessed using Image Lab Software (version 6.0.1, Bio-Rad Laboratories).

### 4.7. Treatment of Cells with Caspase Inhibitors and ROS Scavenger

The caspase peptide inhibitors Z-DEVD-FMK and Ac-YVAD-CMK and the ROS scavenger N-acetyl-L-cysteine were freshly dissolved in culture medium to achieve the final concentrations. SH-SY5Y cells seeded into 96-well plates at a density of 1 × 10^4^ cells per well and then pretreated with Z-DEVD-FMK (20 mM), Ac-YVAD-CMK (20 mM), or N-acetyl-L-cysteine (20 mM) for 1 h. After washing the cells with DPBS, cells were exposed to 200 µM CoCl_2_ for 24 h in a 5% CO_2_ humidified atmosphere at 37 °C. The viability of SH-SY5Y cells treated with caspase inhibitors was determined using the CCK-8 kit. Protein levels were detected by Western blotting.

### 4.8. Statistical Analysis

All data are presented as the means ± standard deviation (SD). Statistical analyses were performed using the Student’s *t*-test using GraphPad Prism 6 software (San Diego, CA, USA). A *p* < 0.05 value was considered statistically significant.

## Figures and Tables

**Figure 1 ijms-25-02185-f001:**
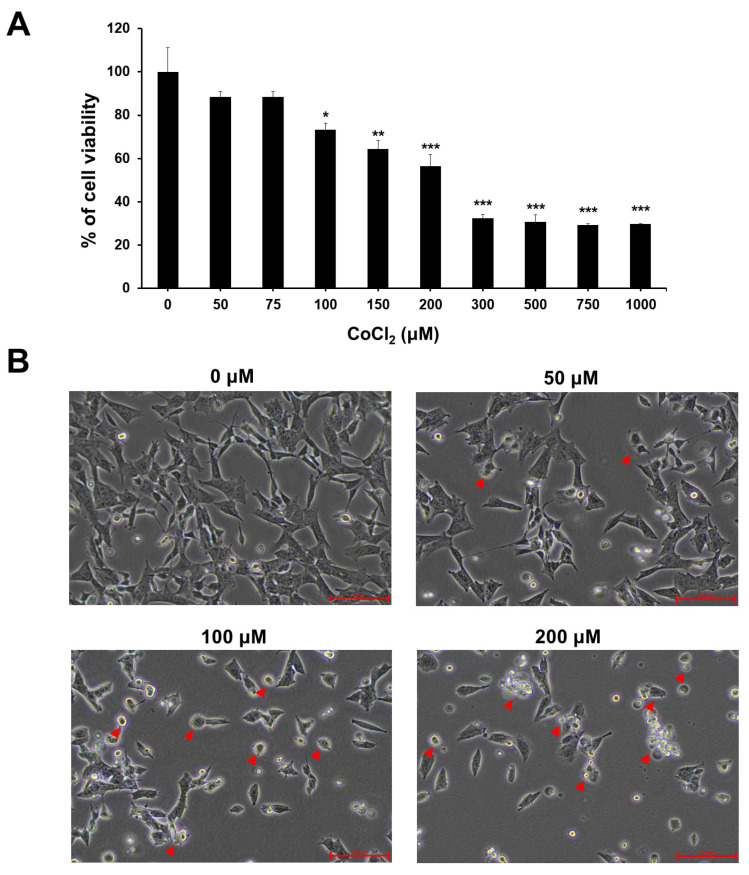
Effect of chemical hypoxia on SH-SY5Y cell proliferation and cellular morphology. (**A**) SH-SY5Y cells were treated with 0−1000 µM CoCl_2_ as indicated for 24 h and detected using a CCK-8 assay. Data are expressed as the percentage of cell viability versus unexposed control cells. Data represent mean ± standard deviation from at least three independent experiments. Statistical analysis was performed by Student’s *t*-test, * *p* < 0.05, ** *p* < 0.01, *** *p* < 0.001 vs. untreated control cells. (**B**) Representative bright-field images of SH-SY5Y cells treated with CoCl_2_ (50, 100, or 200 μM) or vehicle for 24 h. Red arrowheads indicate the damaged SH-SY5Y cells. Images were obtained using a routine inverted light microscope at ×400 magnification.

**Figure 2 ijms-25-02185-f002:**
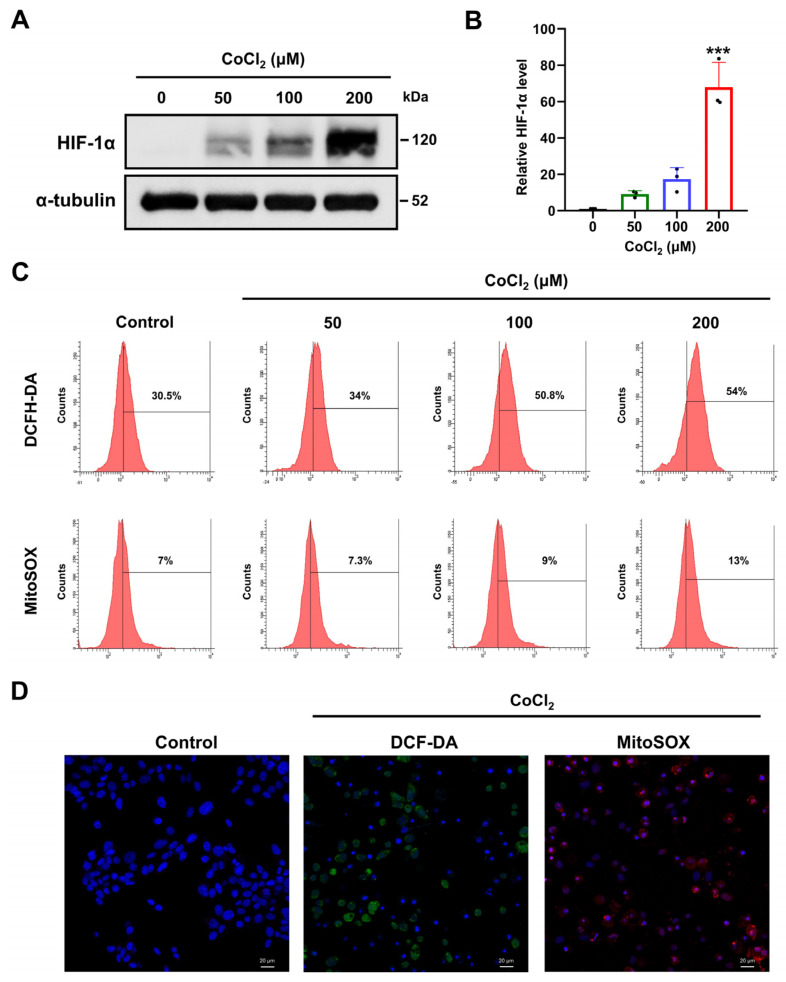
Effect of chemical hypoxia on HIF-1α expression and ROS generation on SH-SY5Y. (**A**) Protein expression levels of HIF-1α were detected by Western blot in SH-SY5Y cells in induced hypoxia with CoCl_2_ (0, 50, 100, or 200 μM) for 24 h. α-tubulin levels are presented as a loading control. (**B**) Relative levels of HIF-1α protein expression following treatment with CoCl_2_. Data are presented as the mean ± standard deviation of three independent experiments, and results normalized to untreated control cells. *** *p* < 0.001 vs. the unexposed control cells according to Student’s *t*-tests. (**C**) Flow cytometric analysis of cytosolic ROS generation using DCFH-DA and mitochondrial superoxide radicals using MitoSOX in SH-SY5Y cells treated with CoCl_2_ (0, 50, 100, or 200 μM) for 24 h. (**D**) Representative fluorescent images of DCFH-DA (green) staining and MitoSOX (red) staining in SH-SY5Y cells treated with 200 μM CoCl_2_ were obtained using confocal microscopy. Non-treated SH-SY5Y cells were used as negative controls. ×200 magnification.

**Figure 3 ijms-25-02185-f003:**
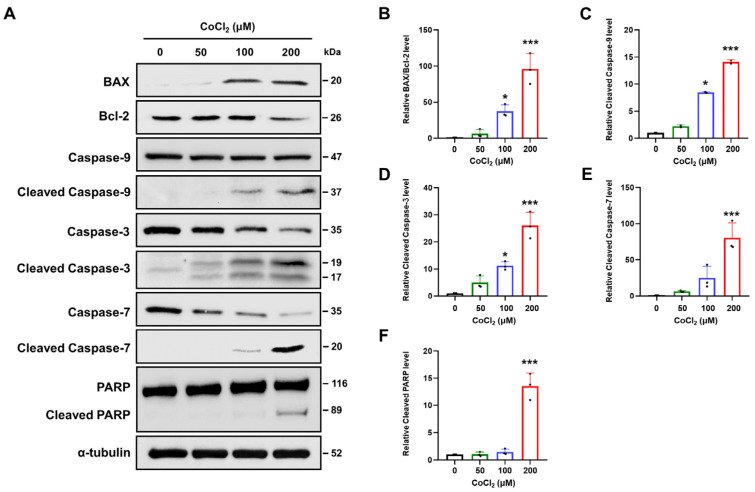
Effect of chemical hypoxia on apoptosis related proteins in CoCl_2_-treated SH-SY5Y cells. Cells were treated with the indicated concentrations (0, 50, 100, or 200 μM) of CoCl_2_ for 24 h. (**A**) Western blotting was used to observe Bax, Bcl-2, caspase-3, cleaved caspase-3, caspase-7, cleaved caspase-7, caspase-9, cleaved caspase-9, PARP, and cleaved PARP protein expression in SH-SY5Y cells. (**B**) The relative gray values of Bax and Bcl-2 were quantified. The Bax/Bcl-2 ratio was calculated, and α-tubulin was used as the internal protein loading control. (**C**–**F**) The relative intensities of the proteins were calculated by comparing them to the intensity of α-tubulin. Data are presented as the mean ± standard deviation of three independent experiments and the results are normalized to untreated control cells. Statistical analysis was performed by Student’s *t*-test, * *p* < 0.05, *** *p* < 0.001 vs. untreated control cells.

**Figure 4 ijms-25-02185-f004:**
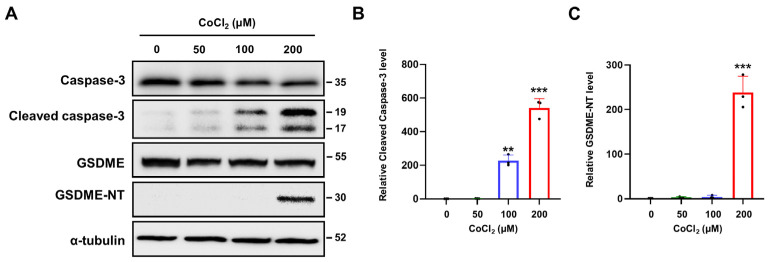
Chemical hypoxia triggers GSDME-mediated pyroptosis in SH-SY5Y cells. (**A**) Protein expression levels of caspase-3, cleaved caspase-3, GSDME, and GSDME-NT were detected using Western blotting analysis in SH-SY5Y cells treated with CoCl_2_ at different concentrations (0, 50, 100, or 200 μM) for 24 h, and the relative protein expressions of (**B**) cleaved caspase-3 and (**C**) GSDME-NT were quantified. α-tubulin was used as the internal protein loading control. Data are presented as the mean ± standard deviation of three independent experiments, and the results are normalized to untreated control cells. ** *p* < 0.01, *** *p* < 0.001 vs. untreated control cells using Student’s *t*-tests.

**Figure 5 ijms-25-02185-f005:**
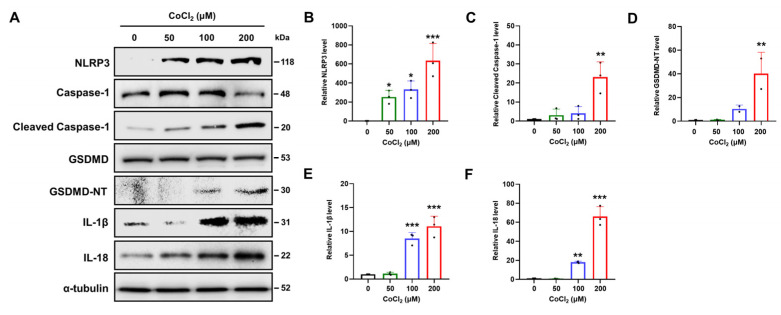
Chemical hypoxia contributes to GSDMD-mediated pyroptosis in SH-SY5Y cells. (**A**) Western blotting was used to observe NLPR3, caspase-1, cleaved caspase-1, GSDMD, GSDME-NT, IL-1β and IL-18 protein expression in SH-SY5Y cells treated with various concentrations of CoCl_2_ (0, 50, 100, or 200 μM) for 24 h. (**B**–**F**) Relative gray values of each group were quantified by densitometry. α-tubulin was used as the internal protein loading control. Data are presented as the mean ± standard deviation of three independent experiments, and the results are normalized to untreated control cells. Statistical analysis was performed by Student’s *t*-test, * *p* < 0.05, ** *p* < 0.01, *** *p* < 0.001 vs. untreated control cells.

**Figure 6 ijms-25-02185-f006:**
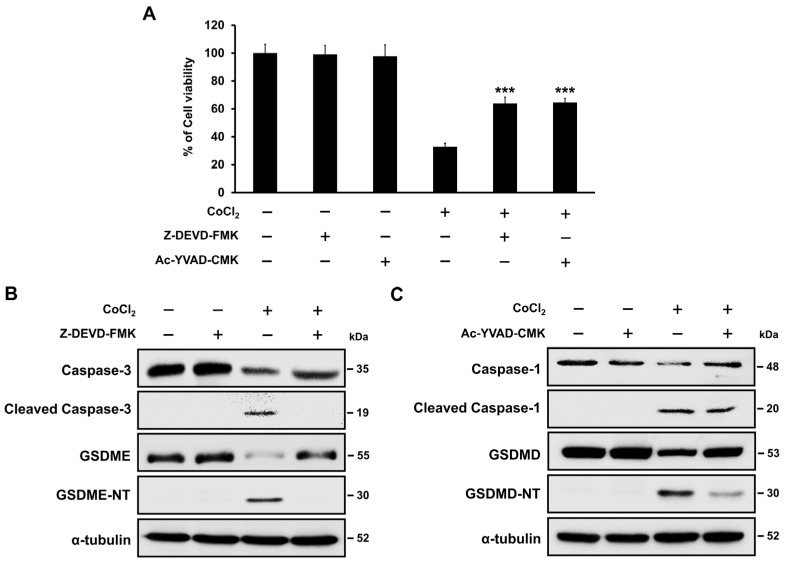
Caspase inhibitors decrease the hypoxia-induced pyroptotic cell death in SH-SY5Y cells. (**A**) Cell viability was analyzed in SH-SY5Y cells after incubation with caspase-3 inhibitor (Z-DEVD-FMK) or caspase-1 inhibitor (Ac-YVAD-CMK) in the presence of 200 μM CoCl_2_ using a CCK-8 assay. Non-treated SH-SY5Y cells were used as negative controls. Data are presented as the mean ± standard deviation of three independent experiments. *** *p* < 0.001 vs. CoCl_2_-treated cells using Student’s *t*-tests. (**B**) Western blot analysis of caspase-3, cleaved caspase-3, GSDME, and GSDME-NT in 200 μM CoCl_2_-treated SH-SY5Y cells after 20 mM Z-DEVD-FMK pretreatment. (**C**) The expression of caspase-1, cleaved caspase-1, GSDMD, and GSDMD-NT in 200 μM CoCl_2_-treated SH-SY5Y cells following pretreatment with 20 mM Ac-YVAD-CMK was determined using Western blot analysis. α-tubulin was used as the internal protein loading control.

**Figure 7 ijms-25-02185-f007:**
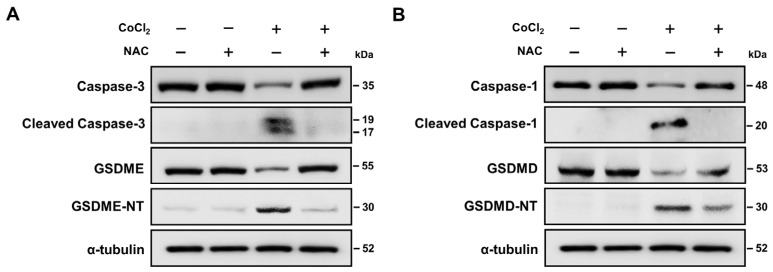
Effects of ROS scavenger on hypoxia-induced pyroptosis in SH-SY5Y cells. (**A**,**B**) Representative Western blot analysis of caspase-3, cleaved caspase-3, GSDME, GSDME-NT, caspase-1, cleaved caspase-1, GSDMD, and GSDMD-NT in 200 μM CoCl_2_-treated SH-SY5Y cells before, or after, 20 mM N-acetyl-L-cysteine (NAC) pretreatment. α-tubulin was used as the loading control.

**Figure 8 ijms-25-02185-f008:**
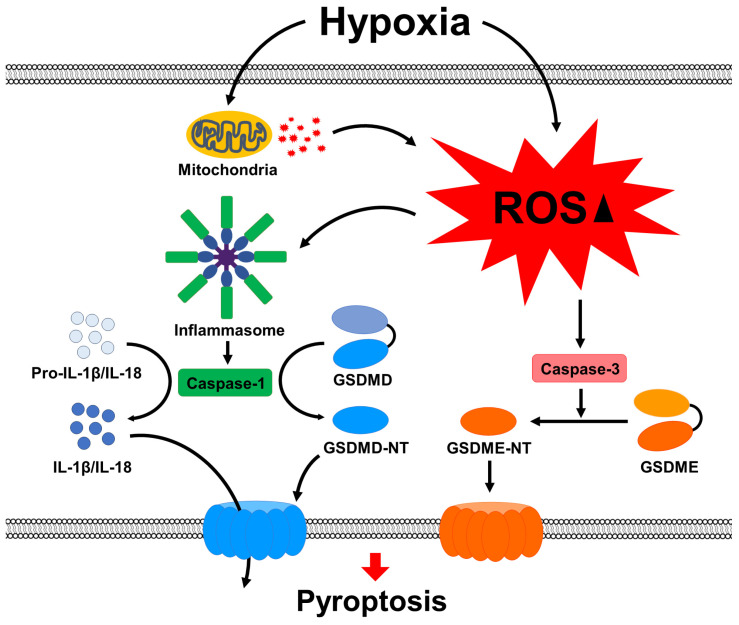
Mechanisms of ROS-mediated pyroptosis under hypoxic conditions.

## Data Availability

The data presented in this study are available on request from the corresponding author.

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
