# Peer review of "Chemical Hypoxia Induces Pyroptosis in Neuronal Cells by Caspase-Dependent Gasdermin Activation"

_ijms, 2024, doi:10.3390/ijms25042185_

Round 1

Reviewer 1 Report

Comments and Suggestions for Authors

Reviewer comments and suggestions

The authors in this study experimentally demonstrated that hypoxia-induced reactive oxygen species (ROS) might trigger pyroptosis via caspase-dependent gasdermin (GSDM) activation in neuronal cells. 

For the study hypothesis, they treated SH-SY5Y neuronal cells to cobalt chloride (CoCl2) to trigger hypoxia and then evaluated cellular and molecular responses to hypoxic conditions. Exposure to CoCl2 elicits excessive accumulation of cytosolic and mitochondrial ROS in SH-SY5Y cells. In SH-SY5Y cells, CoCl2-induced hypoxia not only triggered the intrinsic (caspases-3, -7, and -9) apoptotic pathway but also triggered pyroptosis that was NLRP3/caspase-1/GSDMD-mediated and caspase-3/GSDME-dependent. With the help of selective inhibitors and scavengers the authors concluded that hypoxia-mediated ROS production plays an important role in the activation of both apoptosis and pyroptosis in SH-SY5Y neuronal cells. 

Overall, the manuscript was well written. However, a few major concerns or comments needed to be explained or modified.

  1. Lines 37-38 I think one reference was not enough to highlight all. please add up more here
  2. Line 77 Recent studies have reported that pyroptosis can be induced by hypoxia ( the authors need to add appropriate reference)
  3. Line 80-81 This happens only in cancer cells; what about normal cells
  4. Line 186 How the authors confirm about pyroptosis, please explain here for the common reader of your manuscript
  5. It would be nice if the authors could draw a flow chart of mechanism involvement in the discussion section
  6. Line 278-285 There is no need to mention these lines in the first paragraph of the discussion; directly mention the novelty of this study
  7. Line 318-319 Please explain it
  8. Line 327-328 Please mention the figure number here
  9. Did the authors find any limitation in the study, that can also be acknowledged

Author Response

Comments 1: Lines 37-38 I think one reference was not enough to highlight all. please add up more here

Response 1: Thank you for your suggestion. We added more reference according to your suggestion in the revised manuscript. (line 38)

Comments 2: Line 77 Recent studies have reported that pyroptosis can be induced by hypoxia ( the authors need to add appropriate reference)

Response 2: Thank you for your detailed review. We added references according to your suggestion in the revised manuscript. (line 77)

Comments 3: Line 80-81 This happens only in cancer cells; what about normal cells

Response 3: Thank you for your comment. In this study, we conducted an experiment under the hypothesis that hypoxia damages neuronal cells by activating GSDM, especially GSDME and GSDMD. As far as we know, GSDMC was originally identified as a tumor-processing marker in human melanoma. However, GSDMC is expressed in the trachea, spleen and gastrointestinal tract, stomach, large and small intestines, cecum and colon (Reference https://doi.org/10.3892/ijo.2023.5548). GSDMC has not been studied much among GSDM families. Most GSDMC studies are conducted in tumor cells, and as suggested by the researcher, research on how hypoxia affects the expression of GSDMC in normal cells seems to be a good topic. Thank you for your valuable suggestion.

Comments 4: Line 186 How the authors confirm about pyroptosis, please explain here for the common reader of your manuscript

Response 4: Thank you for your comment. GSDME consists of a cytotoxic N-terminal pore-forming domain and an autoinhibitory C-terminal domain. After GSDME activation, N-terminal domain is liberated and free to oligomerize and insert into the plasma membrane, resulting in pore formation. And pore formation results in loss of osmotic homeostasis, swelling of the cell, and death. It is the key step of the pyroptosis that the GSDME is cleaved and GSDME-NT is activated. In Figure 4, when 200 μM of CoCl2 was treated, GSDME-NT was significantly activated, and in the cytotoxicity experiment in Figure 1, more than 50% of cells were treated with 200 μM of CoCl2. However, as the reviewer said, the data on cell death were not presented in Figure 4, so lines 194-195 were revised as follows: “These results suggested that CoCl2 induced GSDME-dependent pyroptosis through caspase-3 activation.”

Comments 5: It would be nice if the authors could draw a flow chart of mechanism involvement in the discussion section

Response 5: Thank you for your good suggestion. As per your suggestion, we added new Figure 8 that explains the mechanism in the revised manuscript.

Comments 6: Line 278-285 There is no need to mention these lines in the first paragraph of the discussion; directly mention the novelty of this study

Response 6: Thank you for the insightful comments on the discussion section. As per your suggestion, we removed the line 278-285 in the revised manuscript.

Comments 7: Line 318-319 Please explain it

Response 7: Thank you for your suggestion. It is well known through previous studies that GSDME can be cleaved by activated caspase-3 to generate its N-terminal fragment GSDME-NT, which performs pyroptosis. In Figure 6, it was confirmed that CoCl2 induced chemical hypoxia causes caspase-3 cleavage and activates GSDME-NT, and the cell viability of cells was only about 30%. The caspase-3 inhibitor was treated to confirm that pyroptosis occurred by the activation of caspase-3. GSDME-NT was activated when CoCl2 was treated to induce hypoxia, but it was confirmed that GSDME-NT was not activated when treated with caspase-3 inhibitor despite treatment with CoCl2. In addition, when treating caspase-3 inhibitor, cell visibility also increased by more than 60%. This result means that hypoxia activated caspase-3, and activated caspase-3 induced pyroptotic cell death through the activation of GSDME-NT.

Comments 8: Line 327-328 Please mention the figure number here

Response 8: Thank you for your comment. As per your suggestion, we added figure number in revised version. (line 319)

Comments 9: Did the authors find any limitation in the study, that can also be acknowledged

Response 9: Thank you for your detailed review. The limitation of the present study is that the experiments were conducted using only one cell line. Thus, additional comparative studies are needed using normal brain cells, including primary neurons, microglia, and astrocytes. We added this as a limitation on Discussion in the revised manuscript.

Reviewer 2 Report

Comments and Suggestions for Authors

The authors present an interesting study examining the mechanisms associated with cobalt chloride-induced hypoxia in SH-SY5Y cells as a model of hypoxic disease states. Briefly, SH-SY5Y cells were incubated with cobalt chloride up to 1000 uM for 24 hours and the impact on indices such as cell viability, ROS production, and pro-apoptotic pathways was assessed, predominantly at concentrations up to 200 uM. Increasing concentrations of cobalt chloride was observed to increase cell death through a hypoxic pathway (as confirmed through HIF-1alpha changes), with cell viability determined to be influenced greatly via changes in ROS levels, that in turn influenced pathways associated with caspase dynamics. In total, this study adds to the existing knowledge on the hypoxic response of cells in the context of health and disease, while also identifying potentially new therapeutic targets which warrant further investigation.

In reviewing the manuscript I made a couple of observations. The following should be considered by the authors when preparing a suitable resubmission.

1.       Did the authors perform any experiments in which the influence of the caspase inhibitors and ROS inhibitors alone on indices such as viability and others?

2.       When performing the experiments with caspase inhibitors and ROS inhibitors, were any positive controls run to validate the effect hey were having on the indices being measured?

3.       Were any studies performed to determine if the inhibitors influenced HIF-1alpha levels at all?

Author Response

Comments 1: Did the authors perform any experiments in which the influence of the caspase inhibitors and ROS inhibitors alone on indices such as viability and others?

Response 1: Thank you for your detailed review. In the case of the caspase inhibitor, cell viability was measured when Ac-YVAD-CMK and Z-DEVD-FMK were treated alone, respectively. When Ac-YVAD-CMK and Z-DEVD-FMK were treated alone, there was no significant effect on cell viability. According to the reviewer's opinion, these results were added to Figure 6A in the revised manuscript. In experiments treating ROS inhibitors, cell viability was not measured. However, looking at the Western blot analysis data in Figure 7A, there is no significant difference from control when only the ROS inhibitor is treated alone.

Comments 2: When performing the experiments with caspase inhibitors and ROS inhibitors, were any positive controls run to validate the effect hey were having on the indices being measured?

Response 2: Thank you for your comment. Unfortunately, no experiments on positive control were conducted in caspase inhibitors and ROS inhibitors experiments. However, in the experimental results, it was clearly confirmed that caspases and gasdermins were activated when hypoxia was induced with CoCl2. In addition, it was confirmed that the activities of caspases and gasdermins were reduced when CoCl2 was treated with inhibitors together. Our research team is planning to continue to conduct pyroptosis research, and according to the reviewer’s opinion, we will make sure to use positive control in future studies. Thank you for your valuable suggestion.

Comments 3: Were any studies performed to determine if the inhibitors influenced HIF-1alpha levels at all?

Response 3: Thank you for your suggestion. We only conducted an experiment on whether CoCl2 induces the expression of HIF-1alpha. In this study, we hypothesized that hypoxia-induced ROS may trigger pyroptosis via caspase-dependent gasdermin (GSDM) activation in neuronal cells. Our results showed that CoCl2 elicits excessive accumulation of cytosolic and mitochondrial ROS in SH-SY5Y cells, and CoCl2-induced ROS activated caspase-3/GSDME-dependent and NLRP3/caspase-1/GSDMD-mediated pyroptosis in SH-SY5Y cells. Like the reviewer's opinion, it would have been nice to check HIF-1alpha levels as well, but it was confirmed that HIF-1alpha was strongly expressed when treated with 200 μM CoCl2, and the experiment of the inhibitor focused on changes in proteins related to pyroptosis according to the subject of this study. Thank you for your good opinion and in the next hypoxia experiment, as the reviewer's opinion, we will experiment with the expression of HIF-1alpha after inhibitor treatment.
